# Foodborne Clostridioides Species: Pathogenicity, Virulence and Biocontrol Options

**DOI:** 10.3390/microorganisms11102483

**Published:** 2023-10-03

**Authors:** Mary Garvey

**Affiliations:** 1Department of Life Science, Atlantic Technological University, Ash Lane, F91 YW50 Sligo, Ireland; mary.garvey@atu.ie; 2Centre for Precision Engineering, Materials and Manufacturing Research (PEM), Atlantic Technological University, F91 YW50 Sligo, Ireland

**Keywords:** Clostridioides, foodborne, mortality, resistance, biocontrol

## Abstract

Clostridioides species possess many virulence factors and alarming levels of muti-drug resistance which make them a significant risk to public health safety and a causative agent of livestock disease. Clostridioides result in serious systemic and gastrointestinal diseases such as myonecrosis, colitis, food poisoning and gastroenteritis. As foodborne pathogens, Clostridioides species are associated with significant incidences of morbidity and mortality where the application of broad-spectrum antibiotics predisposes patients to virulent Clostridioides colonisation. As part of the One Health approach, there is an urgent need to eliminate the use of antibiotics in food production to safeguard animals, humans and the environment. Alternative options are warranted to control foodborne pathogens at all stages of food production. Antimicrobial peptides and bacteriophages have demonstrated efficacy against Clostridioides species and may offer antimicrobial biocontrol options. The bacteriocin nisin, for example, has been implemented as a biopreservative for the control of *Listeria*, *Staphylococcus* and *Clostridia* species in food. Bacteriophage preparations have also gained recognition for the antibacterial action against highly virulent bacterial species including foodborne pathogens. Studies are warranted to mitigate the formulation and administration limitations associated with the application of such antimicrobials as biocontrol strategies. This review outlines foodborne Clostridioides species, their virulence factors, and potential biocontrol options for application in food production.

## 1. Introduction 

The Clostridioides (Clostridium) genus comprises Gram-positive spore-forming anaerobes associated with morbidity and high rates of mortality in patients. This robust genus is found in soil, water and the gastrointestinal tract (GIT) of animals and humans (often asymptomatically) and other ecological niches [1]. As an active member of soil ecosystems, some *Clostridioides* species are involved in nitrogen fixation (e.g., *C. pasteurianum*) and phosphate solubilisation enabling plant growth [2], while other members are responsible for plant disease, e.g., *C. bifermentans* and *C. subterminale* infection of kiwi plants [3]. *Clostridioides* species including *C. botulinum*, *C. perfringens* and *C. difficile* have long been established as human pathogens and a major public health risk with poorly understood pathology [4]. The European Society of Clinical Microbiology and Infectious Diseases (ESCMID) publishes classification and treatment guidelines for *Clostridioides* disease prevention and management, as summarised in Table 1. *Clostridioides* colonise the GIT of patients post depletion of the normal intestinal microbiota commonly associated with antibiotic therapy [5]. *Clostridioides* difficile infection (CDI) has been monitored by the European Union (EU) in hospitals since 2016, with a reported 18.3 million cases in 2016–2017 [6] with a 30-day mortality rate of ca. 30%. *C. difficile* is well recognised as a nosocomial pathogen and the causative agent of healthcare-associated infections (HAIs), including antibiotic-associated diarrhoea and antibiotic-associated colitis, including pseudomembranous colitis globally, resulting in gross economic burden of ca. €300 million per year in the EU [7]. Risk factors for Clostridioides infection include age, surgical procedures [5], drug therapy, co-morbidities, renal disease, hepatic disease, immunosuppression and exposure to antibiotics including beta-lactams, fluoroquinolones and clindamycin [8]. Research has also demonstrated a significant rate of community-acquired *C. difficile* infection (41%) in younger persons without past antibiotic exposure [9]. Due to its intestinal location, *C. difficile* is also associated with foodborne transmission via the faecal–oral route via contaminated seafood, vegetables and meat products [10]. While nosocomial transmission of *C. perfringens* does occur, community-acquired infections are more common than HAIs [11]. *C. perfringens* is widely associated with foodborne transmission, with the CDC estimating ca. 1 million *C. perfringens* cases of foodborne illness yearly, worldwide [12] with a 30-day mortality rate of *C. perfringens* bacteraemia of ca. 44% [13]. *C. perfringens* infection is associated with gastroenteritis, skin and soft tissue infection, gas gangrene (myonecrosis), bacteraemia, sepsis, liver abscess, muscle necrosis and intravascular haemolysis [14], amongst other clinical symptoms. Alarmingly, sepsis resultant from *C. perfringens* infection has a mortality rate of 74% in certain patient cohorts [14]. *C. botulinum* results in infant and adult intestinal botulism after intestinal colonisation and production of the botulinum neurotoxin (BoNT) [15]. Clostridioides infection is also associated with frequent disease recurrence (<8 weeks post first occurrence or initial incidence of infection) with ca. 35% of cases reoccurring a second, third and subsequent time [16]. Patients presenting with inflammatory bowel diseases (IBDs), which are autoimmune diseases of the GIT, have higher rates of CDI, more severe clinical symptoms, increased colectomy, and increased cases of recurrence [5]. Clostridioides have many virulence factors and possess a high level of antimicrobial resistance (AMR) which promotes their pathogenicity, transmission, morbidity and significant mortality rate. Indeed, multi-drug resistant (MDR) *C. difficile* strains are increasingly prevalent and are associated with worse prognosis [5]. Systemic infection and organ damage are associated with poor prognosis in cases of Clostridioides sepsis. There is an urgent need to determine additional, alternative treatment or combination therapies for the control of Clostridioides infection. The aim of this review is to highlight Clostridioides species associated with foodborne disease in terms of their virulence, antibiotic stewardship, and biocontrol options to safeguard public health.

## 2. Pathogenicity and Virulence Factors

Clostridioides possess many virulence factors which allow for their survival, transmission and pathogenicity within the food chain (Table 2). While Clostridioides spores and toxins play key roles in disease pathogenesis, vegetative cells possess virulence factors which enable survival, colonisation and reproduction in host species. The development of disease is also dependent on host factors such as age, immunity, intestinal dysbiosis, co-morbidities, and the use of therapeutics, including antibiotics and proton pump inhibitors [21]. Indeed, intestinal dysbiosis is also associated with many neurological and GIT autoimmune conditions [22,23], further increasing the risk of infection in patients with higher rates of recurrence [5]. 

### 2.1. Virulence Factors of Vegetative Cells

Vegetative cells of Clostridioides have many virulence factors that increase their pathogenicity, including biofilm formation, adhesions, capsules, flagella, plasmids, surface proteins (cell wall and surface layer proteins), pili, and fibronectin-binding proteins [41]. Additional virulence factors present in some Clostridioides species include hydrolytic enzymes collagenase, hyaluronidase, and proteases [42]. Collagenase production in *C. perfringens,* for example, degrades collagen which may result in intestinal tissue damage and tissue necrosis in patients [43]. Clostridioides neuraminidases result in hydrolysis of terminal sialic acids in host cell membrane glycoprotein, glycolipids and polysaccharides, promoting Clostridioides attachment to host intestinal cells [43]. Microbial biofilms are associated with increased resistance to antimicrobials and host immunity, while allowing for persistence on biotic and abiotic surfaces associated with food production. Several Clostridioides species, including *C. perfringens* [44], *C. difficile* and *C. botulinum,* produce these robust biofilm communities. They also add to the adhesion and dissemination of this organism in vivo. Movement towards, attachment to, and invasion of host intestinal cells are mediated by the presence of flagella on Clostridioides species [42]. While *C. perfringens* species lack flagella, they possess type IV pili, giving them gliding motility aiding in biofilm adherence [45]. Adhesion to epithelial cells in vivo is a pre-requisite to colonisation and is achieved via fimbriae and other appendices, where toxin production subsequently commences [44]. Surface layer proteins (SLP) are adhesion factors that aid in the adherence of *C. difficile* to the colon wall, with bacterial fimbriae aiding in the attachment to the intestinal mucosa and the delivery of toxins to enterocytes [42]. The capsule-like layer and cell wall glycopolymers, including capsular polysaccharides and teichoic acids, provide protection, AMR, host adhesion and immune evasion in *Clostridioides* species [46]. Plasmids are ubiquitous in bacteria including Clostridioides and confer additional genetic traits including toxin genes and AMR genes, increasing pathogenesis. *C. perfringens,* for example, can carry up to ten plasmids, with certain strains having three toxin plasmids and a single plasmid having up to three toxin genes [24]. *C. perfringens* isolates with a chromosomal-located CPE gene have a competitive advantage over isolates with a plasmid-located CPE gene, and are more resistant to several food preservation procedures including heating, refrigeration and freezing [33]. Plasmid pMETRO in *C. difficile* confers resistance to the antibiotic metronidazole [39]. Alarmingly, the sharing of BoNTs genes on plasmids has allowed other clostridial species to produce *C. botulinum* neurotoxins [47]. The presence of toxin genes on plasmids facilitates horizontal gene transfer (HGT), genetic recombination and the emergence of new virulent subtypes [48]. 

### 2.2. Clostridioides Spores

Clostridioides spores are capable of surviving the unfavourable conditions used in food preservation methods, including temperature fluctuation, pH variations and the presence of oxygen, enabling the survival of the bacteria in the food chain and environment [30]. Spores then germinate in favourable conditions, e.g., *C. perfringens* spores germinate at temperatures ranging from 15 to 55 °C [32]. Types A and C *C. perfringens* foodborne illness are resultant from improper cooking or storage of foods, allowing for the survival of resistant *C. perfringens* spores which germinate and cause illness post consumption [33]. The *C. perfringens* enterotoxin (CPE) is produced by this species during sporulation, and is causative of *C. perfringens* food poisoning and associated diarrhoea [43]. Once the vegetative cell has completed sporulation, the cell lyses and releases the CPE into the intestinal lumen [49]. Type A *C. perfringens* causes myonecrosis when spores infiltrate muscle tissue and germinate, allowing vegetative cells to proliferate, producing alpha toxin and perfringolysin O, causing necrosis, systemic dissemination, organ damage and morbidity [33]. *C. perfringens* is mesophilic and can grow at temperatures from 20 to 53 °C, with *C. perfringens* spores surviving temperatures of 95 °C for 1 h [50]. Studies report the survival of *C. difficile* spores at freezing temperatures (−4 to −20 °C) for ca. 2 months, with 4 months’ survival of spores in meat samples at room temperature, refrigeration and freezing at −80 °C [51]. *C. difficile* is an obligate anaerobe, and its spores allow it to survive and transmit via the faecal–oral route. As an anaerobe, germination is activated in the ileum post exposure to bile acids that induce germination at suitable pH levels where vegetative *C. difficile* proliferates in the colon [52]. Indeed, 15–35% of cases of recurring CDIs are associated with the presence of these robust spores within the host intestine post antibiotic therapy [53]. The *C. difficile* spore is surrounded by a protein coat termed the exosporium, which contains hair-like projections and is believed to allow for adherence to inert surfaces [54] including stainless steel, allowing for persistence in food and hospital environments [55]. Interestingly, studies have described the pro-inflammatory action of *C. difficile* spore proteins producing an immunogenic effect in the patient [53]. Studies have shown that *C. difficile* spores are cytotoxic to host macrophages [54]. The ability of *C. difficile* to adhere to intestinal epithelium is considered a factor in the pathogenicity of clinical disease. The exosporium is believed to aid in the adherence of *C. difficile* spores to the epithelium and its composition varies amongst strains, suggesting why subvariants have different pathogenicity [56]. *C. difficile* spores can survive temperatures of 71 °C for 2 h, and 85 °C for 10 min, indicating the failure of current pasteurisation or thermal preservation techniques to eliminate *C. difficile* spores from food items [51]. The exosporium is absent in *C. perfringens* (where the outermost layer is the spore coat) [33], but also present in *C. botulinum* where it is believed to confer chemical resistance and protection to the infectious spore [57]. The spores of *C. botulinum* are extremely heat resistant; *C. botulinum* spore inactivation is the target for thermal processing of low-acid canned (pH > 4.6) food items [31]. Additionally, spores can survive milk pasteurisation, and contamination from animal faecal matter raises the risk of their presence in milk and dried milk products. Good manufacturing practice (GMP), temperature control, andeffective sanitation at harvest and post-harvest are required to prevent food contamination. While biocidal resistance has been identified in many AMR bacteria [34], food production biocides such as phenols, alcohols, and quaternary ammonium compounds (QACs) have limited efficacy against bacterial spores at concentrations and exposure conditions used in food production facilities [58].

### 2.3. Toxin Production

Toxin production by Clostridioides presents food processing challenges and public health risk. *C. perfringens* is categorised into seven toxinotypes, types A to G inclusive, based on the production of alpha toxin, enterotoxin (CPE) and necrotising (NetB) toxin, with types A and F causative of sepsis, gas gangrene, enterocolitis, hepatobiliary infection, haemolysis, bacteraemia, and mortality [11]. *C. perfringens* toxins are classified into pore-forming toxins, membrane-damaging and hydrolytic enzymes, and intracellular toxins and are carried on large plasmids [59]. The chromosomally located alpha toxin phospholipase C (plc) is the main virulent factor in all strains of *C. perfringens* associated diseases [60]. The pore-forming, cytotoxic CPE toxin is the main toxin causative of *C. perfringens* food poisoning and non-foodborne diarrhoea, and can result in the necrotising of the human ileal and colonic epithelium cells associated with colitis present in clinical strains [43]. *C. perfringens* type F strains producing the enterotoxin CPE are responsible for ca. 15% of antibiotic-associated diarrhoea cases with high rates of recurrence and high levels of treatment failure [61]. Ingestion of the CPE toxin alone is sufficient to induce intestinal symptoms of diarrhoea, such as cramping, and may induce histologic damage, villus blunting, epithelial necrosis and shedding [33]. The infectious dose of *C. perfringens* CPE-positive cells is low, suggesting that plasmid transfer from the CPE gene to the host *C. perfringens* in the GIT microbiome may facilitate pathogenesis [59]. Type F toxigenic strains are also more adherent to intestinal cells, namely Caco-2, compared to other food poisoning strains [43]. Toxin production in *C. perfringens* is dependent on the presence of toxin encoding genes cpa/plc, cpb, etx, iap/ibp/itx, cpe, and netB genes, coding α, β, ε, ι, CPE, and NetB, respectively (Table 3) [11]. Plasmid-encoded toxins along with chromosomal genes must also be considered to properly determine *C. perfringens* toxin production where CPE, ε-toxin, ι-toxin, NetB, β2-toxin and binary enterotoxin (BEC) have all been detected on *C. perfringens* plasmids [43]. CPE production is associated with sporulation in *C. perfringens* regardless of whether the gene is chromosomal or plasmid located [33]. Clostridial myonecrosis and bacteraemia have a mortality rate of ca. 100% in untreated cases, with surgical treatment and antibiotic therapy reducing this to ca. 50% [62].

*C. difficile* is categorised into eight different groups or clades, containing hypervirulent ribotypes (RT) with distinct clinical, genetic, and microbiological characteristics [63]. Of these clades, clade 3 is least studied and contains RT 023 strains, with the virulent RT 027 in clade 2 and RT 078 belonging to clade 5 [64]. The studies of Rohana et al., (2020) determined that 81.4% of clinical isolates belonged to clade 1, with 14.3% belonging to clade 4, and 4.3% belonging to clade 2 [65], which varies in clinical cases globally. *C. difficile* produces three exotoxins A, B and CDT, where toxins A and B inactivate GTPases and result in *C. difficile*-associated colitis, with ribotypes also producing CDT involved in colitis progression, pseudomembranous colitis, disease recurrence and sepsis [30]. *C. botulinum* pathogenesis is the result of the production of an extremely potent neurotoxin, the botulinum neurotoxin, causing paralysis and even fatality at 30–100 ng [36]. Neurotoxigenic *C. botulinum* strains are also classified according to the type of toxin they produce. Currently, nine types of BT neurotoxin strains have been identified, namely, A, B, C, D, E, F, G, H, and X [31]. Botulinum A toxins are relatively heat sensitive and can be destroyed at pasteurization temperatures (63 °C, 30 min), whereas toxin B appears more resistant to this treatment [66]. BoNT toxin is eradicated by heating to 80 °C for 20 min or 85 °C for 5 min. The optimum temperature for growth and BoNT production of proteolytic strains is ca. 35 °C and ca 28 °C for non-proteolytic strains [67]. Interestingly, nonproteolytic *C. botulinum* types B, E, and F can produce toxins at refrigeration temperatures of 3–4 °C.

**Table 3 microorganisms-11-02483-t003:** Foodborne Clostridioides species classification and toxins produced.

Clostridioides Species	Classification	Toxin Produced
*C. difficile* [63]	Clade I, II, III, V, VI, VII, VIII	A, B, CDT *
	Clade IV	B, CDT
*C. perfringens* [25]	A	CPA, BEC [35]
	B	CPA, CPB, ETX [43]
	C	CPA, CPB, CPE
	D	CPA, ETX, CPE
	E	CPA, ITX, CPE
	F	CPA, CPE
	G	CPA, NeTb
*C. botulinum* [31]	I	Botulism toxin A, B, F
	II	Botulism toxin B, E, F
	III	Botulism toxin C, D

CAP—Alpha-toxin, BEC—Binary enterotoxin of *C. perfringens,* CPB—Beta-toxin, ETX—Epsilon-toxin, ITX—Iota-toxin, CPE—Enterotoxin, NeTB—Necrotic enteritis B-like toxin. * CDT presence can vary amongst subspecies in Clade 1 [65].

## 3. Foodborne Transmission of Human Pathogenic Clostridioides

Clostridioides present in wastewater, wastewater biosolids, soil, and irrigation water result in the contamination of vegetables and filter-feeding seafood [68], with faecal contamination of livestock animals at slaughter resulting in meat contamination [30]. The presence of Clostridioides spores from faecal contamination leads to water and soil contamination, ultimately allowing for ubiquitous dissemination throughout the food chain. The addition of antimicrobial food preservatives, exposure to stress environments (cooking, refrigeration) and a lack of uniform hygiene practices from harvest and post-harvest to food consumption impacts on the microbial load present in food items [32]. 

The foodborne transmission of *C. difficile* has become increasingly evident, although this pathogen was once believed to be primarily nosocomial in nature. The studies of Marcos et al., (2021) for example, isolated *C. difficile* from farm to fork, with concentrations of 4.3, 5.8 and 6.8 log^10^ cfu/g in coleslaw, spinach and cottage cheese, respectively [30]. *C. difficile* has been isolated from food-producing animals, their carcasses, and in food processing facilities as well as in raw, ready to eat (RTE) and cooked foods [69]. *C. difficile* has been detected in meat products, ground beef, pork, turkey, vacuum-packed meat, and various meat sausages [51]. Indeed, *C. difficile* has been detected in 6% of RTE food samples [30]. Currently, there are over 800 ribotypes of *C. difficile*, with RT 027 and RT 078 prevalent in human cases of infection [29]. The hypertoxigenic strain RT 027, which emerged in 2002, produces greater levels of toxins A and B (Table 2), which are causative agents of colitis in patients and are believed to have a higher mortality rate than non-ribotype [56]. Such hypervirulent ribotypes can possess toxins A and B and the binary toxin *C. difficile* transferase (CDT) [28]. The heterogeneous nature of Clostridioides species may be a result of their recombination in vivo, HGT and evolutionary reproductive capacity [32]. Ribotype 027 has been detected in livestock animals including sheep, cattle and poultry [30]. Ribotype 078 has been detected in up to 100% of piglets and 56% of young cattle, with varying distribution globally [70]. The studies of Bacheno et al., (2022) report the presence of MDR *C. difficile* (having resistance to metronidazole, ciprofloxacin, and clindamycin amongst others) including ribotypes 027 and 078 on surface swabs of meat and carcasses where strains also possessed two toxigenic genes, imparting increased virulence and high pathogenicity [71]. The presence of *C. difficile* in meat and RTE food items is a public health risk, as mild cooking is not sufficient to destroy *C. difficile* spores [30]. 

Of the seven subtypes of *C. perfringens*, types A and F are associated with human disease; type A produces the alpha-toxin phospholipase C or CPA, resulting in gas gangrene, hepatobiliary infections, and sepsis, and type F produces CPA and enterotoxins (CPE), causing foodborne illness (Table 3) [11]. *C. perfringens* type A is responsible for the majority of *C. perfringens* foodborne illness, and is often associated with undercooked beef and poultry [35]. Indeed, necrotic enteritis (NE) caused by *C. perfringens* is an issue in boiler hens, leading to increased feed conversion ratios and other welfare and economic burdens [72]. The studies of Bendary et al. (2022) determined that 12.6% of chicken meat harboured *C. perfringens,* with 74% of isolates possessing MDR [32]. The beta toxin coded by the cpb2 gene is the most lethal poultry toxin and has been isolated in many poultry birds that also display MDR [72]. Bhattacharya et al. (2020) reported on an outbreak of enterotoxigenic *C. perfringens* associated with the consumption of reheated cheese sauce [37]. *C. perfringens* has one of the fastest known bacterial doubling times of 8–12 min at 43 °C and 12–17 min at 37 °C [43] giving it a competitive advantage in colonising the GIT, resulting in dysbiosis and pathogenicity. 

Foodborne botulism results from the consumption of the BoNT following the growth of vegetative cells and spore formation in food products. Although *C. botulinum* foodborne illness is rare, the botulism toxin causes a severe form of food poisoning with a high mortality rate at very low levels of toxin concentration [26]. Indeed, *C. botulinum* and its toxins have been detected in food as vegetative and spore forms [31]. Toxin types A and B are associated with *C. botulinum* disease, with soil contamination of vegetables and meat products acting as vehicles of transmission [26]. The botulinum toxin has been detected in vegetables, acid-preserved foods, fish, including canned fish, and processed meats [38]. Studies describe the detection of *C. botulinum* spores in milk where silage or bedding from the animals may have caused contamination [73]. *C. botulinum* has also been isolated in honey samples where ca. 63% were type A strains and ca. 16% were type B [74]. 

The European standard EN ISO 7937 for the detection of *C. perfringens* in food, food production facilities and animal feed is the method for the enumeration of live bacteria in the food chain based on the use of specific growth media [61]. Irrespective of the type of food contaminated with Clostridioides, detecting the species, ribotype and toxins present is an important issue in relation to monitoring Clostridioides foodborne illness [75]. The lack of uniform isolation and detection methodologies for the detection and monitoring of Clostridiodes in foods inhibits accurate comparative analysis across food groups and locations. Currently, detection and identification rely on classical microbiology methods and molecular analysis of toxigenic traits to confirm species and specific ribotypes present in disease outbreaks [68]. The application of real-time polymerase chain reaction (RT-PCR) methods has greatly improved the detection and advancement of research on infectious disease, providing high rates of accuracy in short time frames. For identification purposes, sequencing of the genome, specifically ribosome region S16, is typically performed [2]. Also, the use of PCR for toxinotyping and detection of toxin genes, e.g., *cpe* gene detection assays, can provide insight into contamination and food risk [61]. It is possible, however, that components of the food matrix, e.g., pectin and hemocyanin, may inhibit the successful isolation or PCR amplification of species present in food items [76], leading to false negatives. 

### Antibiotic Treament of Clostridioides Species

Typically, mild cases of *C. perfringens* foodborne illness are self-limiting and are resolved within 24 h. Severe *C. perfringens* infection is often treated with penicillin and clindamycin, a bacteriostatic antibiotic which is included due to its ability to suppress toxin production [25]. For the treatment of *C. perfringens* sepsis and septic shock, penicillin G and clindamycin, tetracycline, or metronidazole in combination with surgical removal may prevent patient mortality [25]. Antibiotics such as streptothricin, streptomycin, tetracycline and sulfapyridine have been used as growth promotors in chicken and pig feed, leading to increased productivity [77]. This application of antibiotics in food-producing animals and in animal feed has encouraged the emergence of MDR strains of *C. perfringens* [32]. Studies have detected *C. perfringens* strains with resistance to tested antibiotics including amoxycillin/clavulanic acid, ciprofloxacin, and norfloxacin in broiler chickens [72]. 

The onset of *C. difficile* infection is associated with antibiotic therapy, with clindamycin, moxifloxacin, and tetracycline often associated with disease recurrence [71]. Currently, metronidazole is no longer considered a first-line antibiotic for adults; vancomycin and fidaxomicin are the therapeutics of choice for *C. difficile* infection where oral administration of non-absorbable antibiotics that target the GIT is required [16]. Intravenous (IV) vancomycin is not suitable for CDI, as it does not enter the colon where infection is present [5]. Fidaxomicin appears better at preventing treatment failure or disease recurrence. Recurrence of CDI symptoms after initial therapy develops in 10–30% of cases, with a third recurrence in ca. 65% of cases [5]. Fidaxomicin inhibits *C. difficile* spores’ formation by adhering to the surface of the spore, reducing the rate of recurrence [52]. In cases of fulminant CDI where hypotension, shock, paralytic ileus, and/or toxic megacolon are present, a high dose of oral vancomycin and IV metronidazole is recommended [20]. The monoclonal antibody, bezlotoxumab, which was approved in 2016, can be used in conjunction with antibiotic therapy for *C. difficile* toxin B types [19]. The use of faecal transplantation for recurring CDI is recommended where treatment failure is evident, in patients with three or more episodes of CDI post initial treatment [20]. Faecal transplantation can also be considered in patients with fulminant CDI with no improvement after 72 h of antibiotic therapy [20]. Faecal transplantation has a cure rate of ca. 93% in recurrent CDI, making it highly successful [78]. In patients who have developed fulminant colitis with progression to systemic toxicity, surgical intervention is considered [5]. MDR strains of *C. difficile* are increasing, with approximately 60% of clinical isolates having resistance to three or more antibiotics [5]. Studies have shown that food isolates of *C. difficile* possess MDR to tetracyclines, macrolides, penems and fluoroquinolones [71]. *C. botulinum* is not currently treated with antibiotic therapy due to the potential for BoNT release post bacterial cell lysis [15]. The treatment for BoNT is an equine anti-BoNT antiserum; however, it is not suitable for infant patients and has a risk of anaphylaxis [15]. Infant patients receive Botulism Immune Globulin Intravenous, BabyBIG, in the United States. Studies have detected cefalotin, trimethoprim-sulfamethoxazole, nalidixic acid, and gentamicin resistance in *C. botulinum* strains [74]. 

## 4. Biocontrol Agents in the Mitigation of Clostridioides

Good manufacturing practices (GMP)- and hazard analysis control points (HACCPs)-based preservation and disinfection techniques are implemented in food production and preparation facilities as microbial hazard control procedures [79]. The control of Clostridioides, however, represents a challenge further proliferated by the emergence of AMR and MDR species. AMR infectious diseases result in extended morbidity, metastatic bacterial infections, disease recurrence, and predispose patients to opportunistic infections [79]. Controlling infectious disease via the stringent application of biocides is effective in preventing nosocomial and foodborne transmission of pathogenic species; such chemicals, however, are also prone to AMR and are not biocompatible [34]. Novel biocontrol options offering a green alternative include antimicrobial peptides and bacteriophages [77]. In line with One Health, eliminating pathogens including Clostridioides from food production environments, and reducing the incidence of animal disease will prevent human disease outbreak and contribute to food sustainability [80]. 

### 4.1. Antimicrobial Peptides against Foodborne Clostridioides

Antimicrobial peptides (AMPs) are peptides consisting of fewer than sixty amino acids produced by prokaryotes (bacteriocins) and eukaryotes such as fungi, insects, animals, humans and plants, having broad-spectrum antimicrobial activity [81]. Additionally, AMPs are a part of the innate immune system, possessing immune-modulating and anti-inflammatory action amongst other beneficial activities [82]. As an over growth of Clostridioides results in enteric inflammation, T cell-mediated proinflammatory immune responses, and ultimately tissue damage, such anti-inflammatory action may offer therapeutic benefits [83]. Bacterial bacteriocins produced by *Bacillus, Lactococcus* and *Enterococcus* species have demonstrated activity against *C. difficile* in vitro [21]. Lantibiotics, namely nisin, which is Food and Drug Administration (FDA) approved for food application, has demonstrated efficacy against MDR species of MRSA, *Enterococcus* and *Clostridium* [84]. Nisin at a concentration of 60–120 g/mL and pediocin provide some anti-sporicidal activity against *C. botulinum* [85]. Nisin is implemented to prevent *C. botulinum* spores in food; research, however, demonstrates varying resistance to nisin in *C. botulinum* strains [85]. Lacticin 3147, produced by *L. lactis,* also has demonstrated anti-sporicidal activity against Clostridioides [85]. The studies of Arthithanyaroj et al. (2021) demonstrated that a peptide hybrid of the insect AMPs cecropin A and melittin has efficacy against *C. difficile,* with a minimum inhibitory concentration (MIC) of 3.9 µg/mL [86]. The combination of lantibiotics Ltnα and Ltnβ, named Lacticin 3147, has demonstrated activity against *C. difficile* amongst other Gram-positive organisms, e.g., MRSA, *Streptococcus* and vancomycin resistant *Enterococcus* (VRE) [87]. The fish AMPs piscidins have potent activity against *C. difficile* aerobically and anaerobically [88]. The human cathelicidin AMP LL-37 has some activity against *C. difficile* at 10 mM; however, the AMP reduced the inflammation and tissue damage present in mice with colitis [82]. Exposure to AMP appears to sensitise *C. difficile* to antibiotics, with R027 being less sensitised than non-hypervirulent strains [88]. The AMP sublancin produced by *Bacillus subtilis* demonstrated activity against *C. perfringens* in broiler chickens [83]. Sublancin also reduced proinflammatory interleukins IL-1β, IL-6, and tumour necrosis factor-α in the mouse intestine, which may alleviate enteric inflammation [83]. Furthermore, sublancin had a positive impact on villus height and crypt ratio in *C. perfringens*-infected mouse models, enabling growth improvement [89]. AMPs NZ2114 and MP1102 derived from the fungal defensin plectasin demonstrated activity against pathogenic *C. perfringens* type A and acted in synergy with antibiotics virginiamycin, aureomycin, bacitracin zinc, lincomycin, and vancomycin [90]. The AMP cLFchimera decreased gut lesions and mortality induced by necrotic enteritis associated with *C. perfringens* in broiler chickens [91]. The AMP A3 reduced the excretion of *C. perfringens* in broiler chickens with improved weight gain [92]. AMPs benefit the animal with antibacterial action on Clostridioides, restoring GIT microbiota and providing anti-inflammatory action [83]. Thus, AMPs improve GIT health, nutrient digestibility and growth in food-producing animals [91]. Therefore, AMPs as feed additives replacing antibiotics in livestock farming may reduce the proliferation of AMR while improving farm economy [92]. As food preservation additives, AMPs have limited impact on the organoleptic and nutritional aspects of the food and are easily digested by the GIT without impact on the microbiome [85]. The impacts of the food matrix and environment on AMP activity however, must be considered, such as protein concentration, fat concentration, and bacterial load [93]. Currently, AMPs have limited application therapeutically due to their formulation limitations, pharmacodynamic and pharmacokinetic issues, chemical instability, and protease degradation in vivo [81]. In food matrices however, AMPs remain stable at varying pH and temperatures, and food processing methodologies [85], and remain potent antimicrobial agents with low host cell toxicity [81].

### 4.2. Bacteriophages against Foodborne Clostridioides

Bacteriophages (phages) are viral obligate intracellular parasites of bacterial cells, where they infect, reproduce and emerge from the bacteria via a lysogenic (temperate) or lytic life cycle [80]. The lytic life cycle is utilised by virulent phages, leading to bacterial cell death upon lysis; temperate phages, however, can implement both the lytic or lysogenic cycle [79]. The temperate cycle allows for the incorporation of viral genetic material into the bacterial chromosome as a prophage, allowing for ongoing phage replication without sacrificing the host [94]. Having either single- or double-stranded DNA or RNA in a protein capsid, phages are abundant in nature. As potent antibacterial agents, phages and phage-derived enzymes termed endolysins have potential as biocontrol agents in food production, as summarised elsewhere [77,95,96]. The International Committee for Taxonomy of Viruses (ICTV) classifies nineteen phage families, with families Myoviridae and Siphoviridae phages of *C. difficile* [96]. Many *C. difficile* phages have been identified, as described elsewhere [97]. *C. difficile* phages utilise a temperate life cycle [96], allowing for survival in unfavourable conditions [94]. Temperate phages may promote pathogenicity in bacterial species by inserting virulence genes such as AMR genes, and toxin genes, e.g., CTX gene-coding cholera toxin, Shiga-converting phages associated with haemorrhagic *E. coli,* and botulinum neurotoxins of *Clostridium botulinum* [95]. The phiSemix9P1 phage codes the binary toxin gene in *C. difficile* [96]. Studies have shown phages can code adenosine-diphosphate-ribosyltransferases (ADPRTs) enzymes, promoting adherence and mucosal colonization of *C. difficile* in vivo [94]. Temperate phages therefore, may contribute to and promote virulence in foodborne pathogens including Clostridioides species. At present, there are no *C. difficile* phages implemented for the treatment of CDI [97]. Intralytix Inc, has developed phage products for the control of *Listeria* (ListShieldTM), *E. coli* (EcoShieldTM) and *Salmonella* (SalmoFreshTM) in food production, as approved by the FDA [98]. Lytic phage products PLSV-1TM acts against *Salmonella* and has been approved for veterinary application, with the phage product Intralytix (INT-401TM) applied for the control of *C. perfringens*-induced necrotic enteritis in poultry [80]. Studies on the lytic phage HNo2 reduced the viability of *C. perfringens* by 99% on chicken meat surfaces at 4 °C in 72 h [99]. The phage CPQ1 specific to *C. perfringens* demonstrated activity in chicken meat and milk products and high heat stability at varying pH ranges (4–9) while not possessing any virulence genes, suggesting its suitability in the control of foodborne *C. perfringens* [100]. Adding the *C. perfringens* specific phage φCJ22 to chicken feed reduced the presence of intestinal *C. perfringens* in chickens with reduced incidence of necrotising enteritis (NE) lesions and reduced NE-associated mortality rates [101]. Two lytic phages, P4 and A3, isolated from poultry and pig faeces demonstrated activity against *C. perfringens* alone and in combination with the bacteriocin nisin H [102]. Phages possessing biofilm penetrating ability have action on bacterial cells and may prevent biofilm formation on surfaces in food processing facilities [103]. Phages have the benefit of self-replication, allowing for continued dosage in vivo, limited emergence of resistance and no biocompatibility issues [79]. Phage resistance in bacterial species is associated with the degradation of phage DNA, inhibiting phage attachment to the cell and replication, modification-restriction systems, and the alteration of receptors preventing phage binding [79]. Phages can mutate, however, to overcome such resistance mechanisms. 

Phage-derived enzymes called endolysins/lysins against Clostridioides offer advantages in food production. Endolysins targeting *C. difficile* namely, CD27L, phyCD, CDGCD11, LCD, and CWH have been identified, with CD27L effective against 30 *C. difficile* strains including some R027 strains [97]. These lysin enzymes produced by lytic phages to degrade bacterial cell walls have good efficacy against Gram-positive species, potentially allowing for use as food preservatives or in livestock feed [103]. The *C. difficile* phage phiMMP01 produces a cell wall hydrolase enzyme which prevents spore germination [104]. Furthermore, the lysin CBO1751 demonstrated activity against *C. botulinum* spore germination, inhibiting toxin production [103]. Lysins active against *C. botulinum* have been identified having efficacy at varying pH ranges (pH 6.5–10.5) and possessing salt tolerance, making them suitable for food production applications [103]. The endolysin LysCP28 produced by phage BG3P has activity against A, B, C, and D types of *C. perfringens* in the presence of duck meat and prevented biofilm formation [105]. LysCPAS15 endolysin inhibited *C. perfringens* in milk at 37 °C, with CP25L effective in contaminated turkey meat [105]. Phage endolysins have a narrow host range allowing for specificity while protecting the host microbiota. There are no resistance mechanisms associated with lysins and no toxicity to host cells due to the absence to peptidoglycan [103]. Lytic phages and phage enzymes may offer effective, potent, environmentally friendly biocontrol agents for application in food production. Lysin, for example, can be added as a purified protein to livestock feed and food as a preservative [98]. Limitations associated with the use phages and phage enzymes include, bacterial resistance, specificity, HGT of virulence genes, effective dose, and food production factors, e.g., temperature, pH and food matrices [78]. Large-scale production of phages and enzymes, formulation for oral administration, and GIT stability are hurdles to phage application in food production [79]. In line with One Health phage therapy used prophylactically and metaphylactically in livestock animals pre-harvest, as food preservatives or sanitation agents may reduce the incidence of foodborne Clostridioides disease. A paradigm shift towards environmentally friendly food production resultant from consumer awareness has occurred. As awareness of the health and environmental impacts of pesticide and biocide use has grown, so too has the demand for unprocessed food items free of chemical pesticide exposure. The use of such green biocontrol agents aligns with current consumer demands for safe food production.

## 5. Conclusions

Foodborne illness remains a significant public health risk, with Clostridioides species amongst the virulent foodborne pathogens. *C. difficile* and *C. perfringens* are increasingly associated with resistant foodborne infectious disease, and recurrent cases are frequently leading to difficult to treat morbidity and alarming mortality rates. As the human population continues to grow, there is an increasing demand for food to meet the needs of the population. With the continued expansion of agriculture, livestock and aquaculture farming, the rate of foodborne infectious disease is also proliferating. Current measures taken to prevent infectious disease and crop losses have become unfavourable, highlighting the need for alternative options to safeguard food production. The use of antibiotics in food production, for example, proliferates antimicrobial resistance in livestock animals with environmental transmission and zoonosis. The use of biocontrol agents such as antimicrobial peptides, phages and lysins in food production may reduce the incidence of foodborne disease. Used alone or in combination, phages and AMPS have antimicrobial, antibiofilm, and sporicidal activity. AMPs have additional beneficial activity including anti-inflammatory and immune regulatory functions. AMPs benefit the animal by having antibacterial activity against virulent Clostridioides, restoring GIT microbiota and providing anti-inflammatory action. Research is needed to overcome the current limitations associated with large-scale production, formulation and stability in order to produce sufficient quantities. At present, there are no marketed phage options against *C. difficile* and *C. botulinum* due to the difficulties associated with spore-forming anaerobic species. Furthermore, *C. difficile* phages currently identified are lysogenic in nature, but lytic phages are optimal for application as biocontrol agents due to their cytotoxic effect on target species and self-replicating nature. The phage product Intralytix (INT-401TM) is available for the control of *C. perfringens*-induced necrotic enteritis in poultry. Lysins demonstrating antiClostridioides activity while having salt and pH tolerance have also been identified. In line with One Health, AMPs, phages and lysins may be applied prophylactically and metaphylactically in livestock animals pre-harvest, as food preservatives or sanitation agents to reduce the incidence of foodborne Clostridioides disease and protect public health. 

## Figures and Tables

**Table 1 microorganisms-11-02483-t001:** Classification of *Clostridioides difficile* infection according to ESCMID guidelines and treatment protocols.

Classification	Description	Recommended Treatment	Additional Recommendations	Antibiotic Stewardship
Non-severe	White cell count of ≤15,000 cells/mL, serum creatinine level ≤ 50% above baseline, core body temperature/fever ≤ 38.5 °C. No imaging features of severity [16]	Fidaxomicin or vancomycin 125 mg, 6 hourly for 10 days or metronidazole, 500 mg 8 hourly for 10 days [17]	Identification of CDI by isolation, contact precautions for suspected CDI cases [5] Hand hygiene with soap and water, surface disinfection and environmental cleaning is essential to prevent transmission [5], use of personal protective equipment (PPE) Not using single-use assays for diagnosis [18] ESCMID-recommended diagnostic algorithm [18] Mab bezlotoxumab and antibiotics for treatment of a second or further recurrence of CDI [19] Asymptomatic carriers of *C. difficile* may disseminate spores in the hospital leading to outbreaks [5]	Treatment of CDI relies on nonabsorbable antimicrobial agents administered orally [16] Cure rates of >90% with vancomycin at dosage of >125 mg orally 3–4 times daily for >10 days Oral metronidazole should be limited to the treatment of an initial episode of mild-moderate CDI [5] Metronidazole associated with a substantial number of treatment failures (25%), 25% relapsed within 1–2 months [19] No use of metronidazole for treatment of severe or recurrent CDIs Fidaxomicin is a poorly absorbed macrolide highly active against *C. difficile* with limited activity against other enteric organisms
Severe	Fever, marked leucocytosis (>15 × 10^9^/L), rise in serum creatinine, Additionally, distension of the large intestine, pericolonic fat stranding or colonic wall thickening. Imaging showing features [17]	Fidaxomicin or vancomycin 125 mg, 6 hourly for 10 days [16]		
Severe complicated/fulminant	hypotension, septic shock, elevated serum lactate, ileus, toxic megacolon, bowel perforation or any fulminant course of disease (deterioration of the patient) [20]	Fidaxomicin or vancomycin 125 mg, 6 hourly for 10 days and consider intravenous tigecycline 100 mg, followed by 50 mg 12 hourly [17]		
Fulminant refractory	CDI not responding to recommended CDI antibiotic treatment, i.e., no response after 3–5 days of therapy [17]	Fidaxomicin, Vancomycin, Tigecycline considered, surgery recommended [17]		

**Table 2 microorganisms-11-02483-t002:** Outlining virulence factors associated with Clostridioides infections disease.

Virulence Factor	Example	Clinical Relevance
Toxins	Alpha toxin (CPA), e.g., phospholipase C, lecithinase [14]	*C. perfringens* type A, haemolysis, epatobiliary infections, sepsis and gas gangrene [14], foodborne diarrhoea [11], necrotic enteritis in fowls and piglets [24]
	CPA and enterotoxins [11]	*C. perfringens* type F, food poisoning [11], Food and feed poisoning animals [24]
	Perfringolysin O (PFO) is a pore-forming toxin having synergistic effects with CPA [25]	* C. perfringens * gas gangrene
	Toxin A, toxin B, binary toxin (CDT)	*C. difficile*, colonocyte death and colitis, CDI extra-intestinal effects [5], *C. botulinum* [26]
	Toxins A (enterotoxin) and B (potent cytotoxin) act as glucosyltransferases	Toxigenic *C. difficile* influences colonic tumorigenesis [27]
	CDT affects ADP-ribosyltransferase [5,28], inhibits the protein actin, damaging the cytoskeleton of GIT cells [29]	Induces necrosis in epithelial cells [28]
Spores	Antibiotic resistance, germination in GIT environment [30] Biocidal resistance—survival in food production environments	Germination of *C. difficile* leads to intestinal inflammation, perforation, toxic megacolon and pseudomembranous colitis [29] *C. botulinum* spores are one of the most heat-resistant pathogenic spores [31], exosporium confers biocide resistance, sporulation and germination of *C. botulinum* produces exotoxins e.g., neurotoxins *C. perfringens* sporulation above 75 °C [32] outgrowth in less than 20 min [33]
Biofilm	Bacterial communities attached to biotic and abiotic surfaces promoting [34], HGT and emergence of species subtypes	Host immune evasion, AMR [34] HGT of plasmids
Capsule	Host adhesion and immune evasion, AMR	Cell wall glycopolymers, including capsular polysaccharides and teichoic acids
Proteins	Degradative enzymes, degradation of host proteins	*C. perfringens* proteases (e.g., clostripain), sialidases (neuraminidases), hyaluronidase (mu toxin), collagenase, and endoglycosidases [35].
	Adhesins, attachment to host cell surface	*C. perfringens* collagen adhesion protein (CNA) and fibrinogen-binding proteins FbpA and FbpB [35]
Oxygen	Strict anaerobe	*C. difficile* [27], *C. botulinum*
	Aerobic tolerance	*C. perfringens*
Temperature	Mesophilic growth at 25–45 °C with optimal growth at 35–37 °C, Psychrotrophic, with an optimum growth temperature of 26–30°C [26]	Proteolytic strains of *C. botulinum* producing type A, B, or F toxins are mesophilic. *C. difficile* optimal growth at 30–37 °C Non proteolytic strains of *C. botulinum* producing type B, E, or F, toxins, can reproduce and form BoNTs at temperatures of 3 °C [26] causing flaccid paralysis and fatality [36] *C. perfringens* growth occurs at temperatures of 12–54 °C [37]
pH	Optimal pH for growth and toxin production is 6.5 to 7.5 [24]	*C. botulinum* will not grow in acidic conditions (pH < 4.6), toxins are stable at low pH [38] Sporulation of *C. perfringens* at pH 6–8 in GIT, viable after 3 months at pH 3 and 10
Plasmid	Carrying additional genetic traits in conjunction with chromosomally located elements.	Carrying toxin genes, e.g., CPE gene in type F *C. perfringens* [35] *C. perfringens* plasmids pCW3-like, pCP13-like, and pIP404-like plasmids [35] Plasmids pCD6, pCD630, pO157 in *C. difficile* [39] Carrying AMR genes, e.g., aminoglycoside/linezolid resistance gene cfrC in *C. difficile* [40]

## Data Availability

Not applicable.

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
