# Peer review of "Foodborne Clostridioides Species: Pathogenicity, Virulence and Biocontrol Options"

_microorganisms, 2023, doi:10.3390/microorganisms11102483_

Round 1
Reviewer 1 Report
This review outlines pathogenicity and virulence of foodborne Clostridioides species and potential biocontrol options for application in food production. Here are some comments.
General comments
1. The author stated virulence factors included antimicrobial resistance (line 12 and line 72). Virulence and antimicrobial resistance are two independent concepts.
2. Many literatures were reviewed. Please add the author’s insights.
Specific comments:
1. line 23-24: this sentence did not refer to Clostridioides.
2. line 32: please rewrite this sentence.
3. line 304: please consider whether to retain this subtitle.
4. please note the format of references.
Author Response
This review outlines pathogenicity and virulence of foodborne Clostridioides species and potential biocontrol options for application in food production. Here are some comments.
General comments
- The author stated virulence factors included antimicrobial resistance (line 12 and line 72). Virulence and antimicrobial resistance are two independent concepts.
Response - This statement has been corrected.
- Many literatures were reviewed. Please add the author’s insights.
Response – additional text has been added as requested. See line 484, 494.
Specific comments:
- line 23-24: this sentence did not refer to Clostridioides.
Response - sentence has been removed. Abstract has been corrected.
- line 32: please rewrite this sentence.
Response - this sentence has been rewritten.
- line 304: please consider whether to retain this subtitle.
Response - title has been rephrased.
- please note the format of references.
Response – references have been checked.
Reviewer 2 Report
The topic of this manuscript is an important in control of foodborne illness, and I am interested in this topic.
Pathogenicity and virulence factors of foodborne Clostridium species are well summarized in this manuscript. In addition, the authors suggested several biocontrol options against the species.
This review is worth reading by many microbiologists and clinicians. I recommend that the manuscript be accepted for publication in Microorganisms.
Author Response
Thank you very much for your kind review of this manuscript. The author appreciates your valuable insight.
Round 2
Reviewer 1 Report
The topic of this manuscript is interesting and important. The authors have revised the manuscript carefully and I suggest accepting it.